# Alternative Splicing at the Crossroad of Inflammatory Bowel Diseases and Colitis-Associated Colon Cancer

**DOI:** 10.3390/cancers17020219

**Published:** 2025-01-11

**Authors:** Paulo Matos, Peter Jordan

**Affiliations:** 1Department of Human Genetics, National Institute of Health Dr. Ricardo Jorge, 1649-016 Lisbon, Portugal; 2BioISI—Biosystems & Integrative Sciences Institute, Faculty of Sciences, University of Lisbon, 1749-016 Lisbon, Portugal

**Keywords:** alternative splicing, colitis-associated cancer, interleukin, signalling pathway, splicing factor

## Abstract

Patients with ulcerative colitis (UC) face a higher risk of developing colorectal cancer (CRC) due to chronic inflammation, a known promoter of tumour growth. Here, we review the molecular differences between colitis-associated cancer (CAC) and sporadic CRC, with a focus on “alternative splicing”, a mechanism by which the same gene can produce various protein forms. We explore how inflammation triggers changes in this process, increasing cancer risk for UC patients. The revised data emphasize that additional research into these molecular changes could help identify new biomarkers (molecules that indicate disease progression) and pave the way for innovative treatments targeting these alterations. Such advances would improve outcomes and quality of life for patients while contributing to cancer prevention and care.

## 1. Introduction

Inflammatory bowel diseases (IBDs) are chronic inflammatory disorders and the two main clinical presentations are ulcerative colitis (UC) or Crohn’s disease (CD). Although incompletely understood, a dysregulated response of the mucosal immune system combined with alterations in the intestinal microbiome play a substantial role in the onset of the disease, particularly in genetically predisposed individuals [1,2].

A recent worldwide epidemiologic survey showed that IBD accounted for about 3.3 million cases in 1990 and 4.9 million cases in 2019, representing an increase in prevalence of almost 50% [3,4]. In Western countries, IBD affects between 0.4% and 0.8% of the population, depending on whether the ethnic background is Asian or White, respectively [5,6]. Among these cases, UC is slightly more frequent than CD. IBD patients suffer from significant morbidity related to strictures, fistulas, perforations, infections and cancer.

## 2. Cancer Risk in Patients with Inflammatory Bowel Disease

The risk of developing CRC is increased about 2-fold compared to the general population in Europe, especially for UC patients [7,8]. Usually, UC patients develop CAC at 8–10 years after disease onset [9], and CAC accounts for about 15% of all UC-related death causes [9]. However, besides this overall statistical increase in CAC risk, significant cumulative risk probabilities have been described along disease duration, reaching up to 14–18% in UC patients with a 30-year history of the disease [10,11]. Disease severity and extent and a family history of CRC [12] are further risk-aggravating factors. In CD, there is additional risk for extra-intestinal cancers that may arise in consequence of immunosuppressive therapies used to treat CD. Over the past 3 decades, the risk of dying from CRC has decreased due to improved clinical intervention but the risk of CRC in patients with IBD has not declined significantly [13].

The increased risk of CAC development is the result of chronic inflammation, a well-known tumour-promoting condition [14], but also related to the intrinsic dynamics of the colon mucosa: a tissue with a high rate of continuous cellular turnover in order to assure regular replacement of the epithelial cells exposed at its surface to the gut lumen. Adult stem cells located at the bottom of the crypts generate new cells each day, which then undergo controlled proliferation in the transit-amplifying zone of the crypts. The newly generated cells then differentiate into the mature cell types found in the gut epithelium and have programmed cell-death mechanisms in place to regulate their turn-over. Thus, the constant presence of cell proliferation cues, together with a requirement for tightly regulated cell differentiation and apoptosis, render this tissue environment particularly susceptible for tumorigenesis [15,16], which is further accelerated by chronic inflammation that favours proliferation over differentiation.

On one hand, an inflammatory environment with activated neutrophils and macrophages generates more reactive oxygen species that cause increased DNA damage and can lead to mutational events in key tumour-promoting genes [17]. Indeed, a 2.4-fold increased mutation rate was found when crypt cells isolated from a normal colon were compared to those from an IBD-affected colon, both in UC and CD patients. The mutations include cell-proliferation-associated as well as reactive oxygen species-associated mutational signatures [18,19]. Some somatic mutations were also identified that might contribute to IBD development, but they are apparently different from those that drive progression to CAC [19,20,21].

In addition, cytokines and growth factors are secreted by macrophages and fibroblasts in an inflammatory microenvironment, and these promote the proliferation and survival of cancerous epithelial cells as well as their epithelial–mesenchymal transition, which loosens the tightly regulated cell differentiation and apoptotic programme in colonic mucosal renewal and ultimately leads to the progression to invasiveness of transformed tumour cells [22].

## 3. Specificities of Colitis-Associated Cancer

Although both occur in the colon, recent studies have revealed some notable differences between sporadic CRC and CAC [8,23,24]. Whereas tumours derived from both sporadic CRC and CAC can present either with a chromosomal instability (about 85%) or with a microsatellite-instability phenotype (15%), the conventional adenoma–carcinoma sequence developing from discrete precursor polyps is only observed in sporadic CRC cases. CAC, on the other hand, follows an inflammation–high-grade dysplasia (HGD)–carcinoma sequence, in which entire areas of chronically inflamed mucosa are prone to neoplastic transformation, a process termed field cancerization [25,26].

The underlying differences were confirmed by transcriptome-based tumour subtyping, which showed that CAC samples lack canonical WNT signalling mutations, namely in the APC tumour suppressor gene, and are instead characterized by a more mesenchymal gene expression profile akin to inflammation-induced dedifferentiation reprogramming [27,28,29]. In consequence, the previously defined consensus molecular subtypes are encountered in a different frequency: CMS2, the WNT-driven subtype, is rare in CAC, while the mesenchymal CMS4 subtype with the activation of TGF-β signalling and the epithelial–mesenchymal transition is highly enriched compared to sporadic CRC. The CMS4 subtype is further characterized by an immunosuppressive, tumour-promoting microenvironment enriched in CD4^+^ regulatory T cells.

Molecular studies further indicated that CAC presents with a different timing and frequency of genetic alterations when compared to sporadic CRC. In particular, gain-of-function missense mutations of the TP53 tumour suppressor gene are found early during CAC carcinogenesis, are more frequent, and occur at different codons compared to sporadic CRC [30]. In particular, few mutations were observed at the mutational hotspots R273, R248, G245, and R175, which characterize sporadic CRC, while common mutations in CAC affect codons R282, R158, H179, or R342, rarely found in sporadic CRC. Notwithstanding, it has been shown that both gain-of-function missense TP53 mutations and loss of wild-type TP53 can contribute to accelerating late-stage CRC progression by activating both oncogenic and inflammatory pathways [31].

In contrast to the early occurrence of TP53 mutations, loss of the APC gene, or silencing of the alternative RNF43 gene from the same WNT pathway, is a late and less frequent event in the dysplasia–carcinoma sequence [27]. CAC development was also associated with extensive copy number alteration, including chromosomal abnormalities, at the transition from low- to high-grade dysplasia [18]. In contrast, mutually exclusive mutations in *KRAS* or *BRAF* oncogenes are found in about 30 or 5% of CAC tumours, respectively [23,26], comparable to their prevalence in sporadic CRC cases.

CRC and CAC development are further exacerbated by the presence of pathogenic microbes in the gut [23,26,32], which were found to be enriched in IBD-patient-derived microbiomes [33]. Once endogenous (individual immune response) or exogenous factors (diet or environmental contaminants) initiate a decrease in the intestinal barrier function, pathogens can infect epithelial cells, cause additional inflammation, and secrete carcinogenesis-promoting metabolites or toxins [34,35].

## 4. Molecular Pathways in Colitis-Associated Cancer

As mentioned above, dysplasia is a characteristic stage during CAC development and reflects the abnormal tissue architecture of the inflamed colon mucosa. Recurring cycles of tissue destruction and tissue repair attempts, which involve cell proliferation but under impaired cell differentiation, precede malignant transformation [36]. These cellular processes involve key inflammation-associated signalling pathways, in both the immune and epithelial cells, particularly the NF-κB and interleukin (IL)-6 pathways [37].

### 4.1. NF-κB Activation

In immune cells, the activation of the transcription factor NF-κB, particularly through the canonical pathway [38,39], is considered a major regulator of an inflammatory response. Upon activation, the DNA-binding subunits of NF-kB bind to the promoters of genes encoding pro-inflammatory cytokines and induces their expression, including TNF-α, IL-1β, and IL-6, but also of IL-12 and IL-23 that are required for the differentiation of T-helper cells, which are abundant in IBD [40,41].

In addition, the cytokine-stimulated NF-κB activation also occurs in epithelial cells that express cytokine receptors at their cell surface. Here, this pathway is mainly involved in cell survival, and genetic interference with its activation in intestinal cells showed that this pathway directly promotes the progression of cancer-initiated cells [42,43,44,45]. The cytokine levels in a chronically inflamed microenvironment maintain NF-κB in its active state.

Subsequent studies also identified that during CAC development, the NF-κB activation can be further sustained by the presence of missense gain-of-function mutations in the TP53 gene [46]. This probably occurs through direct interactions in the cell nucleus between mutant p53 and NF-κB protein subunits and increased promoter accessibility via chromatin remodelling [31]. As referred to above, CAC tumours present with TP53 gain-of-function mutations that occur in the early stages of dysplastic mucosa in consequence of IBD. In contrast, tumours from sporadic CRC are characterized by TP53 missense mutations and loss-of-function in later stages of tumorigenesis.

### 4.2. IL-6/STAT3 Signalling

Multiple studies have shown increased circulating IL-6 levels in CD and UC patients with active disease (but not when in remission) [47,48] and in colon cancer patients [49,50]. IL-6 is secreted by innate immune cells in response to injury to stimulate the survival and proliferation of intestinal epithelial cells, as part of a tissue repair response. Due to this role of IL-6 in the maintenance of intestinal homeostasis, clinical trials with anti-IL-6 antibody therapies revealed limited benefits and several patients exhibited adverse gastrointestinal effects [47,51]. In the presence of chronic inflammation, the pro-proliferative effect of IL-6 also plays a crucial role in the pathogenesis and progression of CAC [52,53,54].

In the colon, IL-6 binds to cell surface-bound or soluble receptors (IL-6R), which then form a complex with coreceptor gp130 [55] and lead to the activation of cytoplasmic protein tyrosine kinases of the Janus kinase family (JAK). Classic IL-6 signalling occurs through the membrane-bound IL-6 receptor, both in immune and epithelial cells, and has a general anti-inflammatory effect to allow for regenerative processes. In addition, a soluble IL-6 receptor fragment (sIL-6R) mediates IL-6 trans-signalling in endothelial cells, fibroblasts, or immune cells that lack the transmembrane IL-6R and generally has a pro-inflammatory effect [56,57]. sIL-6R is an exception as a soluble receptor, because it does not act as a decoy to quench IL-6, but actively promotes cytokine signalling when the complex binds to the co-receptor gp130 on cells. IL-6 trans-signalling was, however, also found as a crucial pathway activator in colon epithelial cells during the development of colitis-associated cancer in murine models [53]. The authors observed that membrane-bound IL-6R expression was suppressed in the epithelial cells, while protein and mRNA levels of membrane-bound gp130 were increased and responded to IL-6/sIL-6R complexes secreted by macrophages, indicating a switch to trans-signalling. Indeed, several studies detected increased serum concentrations of IL-6 and sIL-6R, which correlated with the clinical activity of IBD [58,59].

Well-studied downstream substrates and effector pathways of IL-6 are the signal transducer and activator of transcription-3 (STAT3), the adaptor SHP2/GRB2 complex that leads to RAS/extracellular signal-regulated kinase (ERK), and phosphatidyl inositol 3 kinase (PI3K) for AKT pathway activation [60,61]. All these pathways are well-described oncogenic drivers in the intestinal mucosa [62].

As one crucial step for the malignant transformation of colon cells, the JAK-mediated phosphorylation of the transcription STAT3 activates its binding to specific gene promoter sequences, leading to the transcription of genes regulating cellular proliferation (cyclin D1, proliferating cell nuclear antigen), survival (BCL-xL, survivin), and angiogenesis (VEGF) [63,64,65]. Phosphorylated STAT3 also forms a protein complex with p300 and the RelA DNA-binding subunit of NF-κB that increases nuclear RelA retention for persistent NF-κB activation [66]. Using the immunohistochemical detection of phosphorylated STAT3, patients with active UC had three times more p-STAT3-positive epithelial cells than patients in a disease remission phase. A clear increase in p-STAT3-positive staining also distinguished dysplasia and colitis-associated cancer from control samples [52].

In addition, IL-6-induced STAT-3 activation occurs in tumour-infiltrating immune cells, where it exerts a negative regulatory effect on neutrophils, natural killer (NK) cells, and effector T cells, contributing to an immunosuppressive tumour microenvironment [60].

### 4.3. WNT Pathway

Activation of canonical WNT signalling is an archetypic molecular event in the pathogenesis of sporadic CRC. WNTs are secreted glycoproteins and recognized by the G-protein-coupled receptor Frizzled (FZD) and its co-receptor LRP5/6. This leads to recruitment of the dishevelled protein and inactivation of a cytoplasmic destruction complex, containing GSK-3β, Axin2, and APC, which normally promotes the degradation of cytoplasmic β-catenin [67]. In sporadic CRC, APC mutations compromise this complex, resulting in β-catenin accumulation and its translocation into the nucleus. There, it acts as a co-factor of the TCF and LEF family members of transcription factors, which stimulate cell proliferation and inhibit cell differentiation. In some CRC cases, alternative mutations occur in the *RNF43* gene encoding a transmembrane E3 ubiquitin ligase that acts as a feedback suppressor of the WNT pathway by promoting the degradation of the FZD receptor [68]. Although APC and RNF43 mutations are rare or occur late in CAC [27], there is evidence that alternative mechanisms of WNT pathway activation exist in the progression from IBD to dysplasia and CAC [69]. Early studies suggested that the silencing of genes from the WNT signalling pathway is seen in patients with IBD, with a progressive increase in gene promoter methylation during the development of CAC [70,71]. Increased nuclear β-catenin immunostaining was described in UC patients, indicating WNT-pathway activation [72], but this was not observed in another study, although when CAC-derived organoids were studied, they were able to grow without exogenous WNT supplementation [29]. Alternatively, extensive cross-talk has been documented between the NF-κB and WNT pathway that can lead to cytokine-driven β-catenin stabilization in the cytosol [73].

### 4.4. Other Pathways

The presence of pro-inflammatory cytokines also induces the expression of cyclooxygenase (COX2) and subsequent production of prostaglandins in immune cells. These increase vasodilatation for the recruitment of more inflammatory cells, thus incrementing the inflammatory state, but also bind to G-protein-coupled receptors on colon cancer cells and stimulate their survival through the NF-κB pathway [74].

One cytokine induced by activated NF-κB in immune cells is IL-23, to which a group of T helper cells, Th17, responds specifically with secretion of the IL-17 family of cytokines (IL-17, -21, and -22). Their role in anti-pathogen defence comes at the expense of a perpetuated inflammatory and pro-proliferative environment [75]. The presence of these Th17 in later CRC stages is correlated with a poor prognosis [76,77].

Altogether, the above-described characteristics of CAC are schematically depicted in Figure 1.

## 5. Transcriptomic Reprogramming Through Alternative Splicing

The development of the RNA sequencing technology (RNA-seq), together with the corresponding bioinformatics tools for data analysis, has enabled a deeper look into the transcriptome level, i.e., the complete set of RNA transcripts produced at a given time by the genes expressed in colon cells.

For protein-coding genes, active transcription by RNA polymerase II yields as the first product a pre-messenger RNA (mRNA) that needs to be processed into a mature mRNA. One important processing step is the removal of non-coding introns and correct joining of the coding exon regions of the pre-mRNA. This constitutive splicing process is performed by the spliceosome, composed of ribonucleoprotein complexes and multiple auxiliary proteins, which precisely recognizes the splice sites and catalyzes splicing in a two-step reaction. The composition of this complex and its catalytic activity have been previously reviewed in great detail [78,79,80].

Over 95% of human genes can use the pre-mRNA splicing process to generate more than one mature mRNA isoform from their primary transcript [81]. This process is called alternative splicing and, in most cases, involves the inclusion or skipping of certain exons, accounting for approximately 40% of all alternative splicing events. Other, less frequent variants result from alternative 5′ or 3′ splice site choices (around 25%), intron retention (15–20%), mutually exclusive exons, or alternative promotor polyadenylation site usage (5–10%) [82].

Alternative splicing is not a stochastic process but rather a highly regulated, often tissue-specific mechanism, resulting in two or more mRNA transcripts derived from the same gene. Some splicing isoforms are generated on purpose to contain premature stop codons and become degraded by the nonsense-mediated decay (NMD) pathway of mRNA surveillance. In these cases, most alternative transcripts are characterized by intron retention that leads to a frame-shift and a premature translation termination codon [83]. Examples include the post-transcriptional downregulation of the expression levels of tumour-suppressor genes, including *TP53*, *PTEN*, *CDKN2A,* or *CDH1* [84].

In contrast, many other alternative splicing isoforms are translated into protein. Because they lack or include exon-encoded protein domains, they generate a protein with different functional properties and contribute to the diversity of the human proteome. Numerous examples of such alternative splicing-derived protein variants with different functional or regulatory properties have been identified and reviewed, particularly regarding their role in cancer development [85,86,87,88]. With regard to colorectal cancer, well-characterized splicing variants were recently reviewed [89].

## 6. Exon Definition Governs Alternative Splicing Decisions

The process of pre-mRNA splicing occurs co-transcriptionally and accurately removes introns and ligates exons together to generate the mature mRNA. For this, core spliceosome components recognize conserved sequence elements at the intron–exon boundaries: the U1 small nuclear ribonucleoprotein (snRNP) binds to complementary sequence elements at the 5′ splice site, and U2 snRNP binds to the branch point upstream of the 3′ splice site. When introns are short (i.e., under 200 nucleotides), direct interactions between the spliceosome components bound to the 5′ and 3′ splice sites can occur and are sufficient for correct splicing. In most human genes, however, much longer introns exist between exons that, in turn, are typically short (on average around 150 nucleotides). Therefore, a fundamental general mechanism designated as exon definition exists to ensure correct pre-mRNA splicing, but also to regulate alternative splicing [90].

Exon definition is the process through which the spliceosome and associated factors recognize exons, enabling the binding of U1 and U2-snRNPs to the exon–intron boundaries. Exon definition is based on specific sequence motifs that are recognized by auxiliary splicing factors that can either promote or repress splice site recognition of the respective exon. These motifs are typically six nucleotides long and work as Exonic Splicing Enhancers (ESEs) or Silencers (ESSs) [91]. These elements can also be located in the adjacent introns (as ISE or ISS). Enhancer or Silencer sequence elements are recognized by splicing regulatory proteins, such as members of the serine/arginine-rich (SR) protein [92], or the heterogeneous nuclear ribonucleoprotein (hnRNP) families [93,94]. SR proteins frequently bind to ESEs to promote splicing, while hnRNPs typically bind to ESSs or ISSs to inhibit splicing. They work by mediating or preventing protein–protein interactions with core spliceosome components, and thus either promote or interfere with splice site recognition and spliceosome assembly on a given exon [95], as depicted in Figure 2. These cross-exon protein recognition complexes stabilize spliceosome binding to neighbouring exons, even if separated by the presence of long introns. Over 150 proteins encoded in the human genome can contribute to the regulation of alternative splicing [96,97,98], and many belong to the hnRNP or SR protein families.

Exon definition is crucial for the correct inclusion of most constitutive exons, and even more tightly regulated in the context of alternative splicing. For example, many alternatively spliced exons show weak consensus splice sites, leading to suboptimal recognition by the spliceosome. Thus, their inclusion depends on the support from regulatory splicing factors, the presence of which is orchestrated in response to specific cellular signals [99], or by their expression in a tissue-specific manner [100,101,102]. These regulatory splicing factors operate in a combinatorial manner [95,103,104], as exons contain more than one splice enhancer or silencer sequence element. An additional mechanism contributing to the regulation of alternative splicing events depends on transcription elongation speed, which can be regulated by phosphorylation events in the C-terminal domain of RNA polymerase II, but also depends on chromatin modifications such as histone H3 lysine 36 trimethylation [105,106]: the slower transcription proceeds, the higher the probability that exons with weaker splice site consensus conservation are recognized by the spliceosome and included into the mature mRNA [107].

## 7. The Role of Alternative Splicing in CAC Development

Transcriptomic changes in genes involved in intestinal epithelial homeostasis can decrease the epithelial barrier function, leading to increased exposure to pathogenic microbes or luminal toxins, and this perpetuates inflammation during disease progression or recurrence. Among the transcriptome changes, alternative splicing is one particular aspect retrieved from these studies (apart from changes in micro-RNAs or long non-coding RNAs, which are outside the scope of this review).

Alternative splicing changes in an inflamed colon can be caused by altered cell signalling in response to inflammation, leading to an altered expression, localization, or activity of splicing factors [99,108,109]. On top of this, some changes result from gene variants predisposing individual patients by affecting the ratio of splicing variants of disease-related genes, or the expression levels of splicing factor-encoding genes [110,111], including their epigenetic modulation [112].

In the following, we review examples of deregulated alternative splicing variants or splicing factors in colon epithelial cells and their role in the transition from IBD to CAC. A more comprehensive list of colon cancer-associated splicing variants is summarized in Table 1.

### 7.1. Splicing Variants Associated with CAC

Persistent colon inflammation, driven by pro-inflammatory cytokines, was found to change the alternative splicing of 3560 genes in long-duration UC patient samples, predicting significant alterations in metabolic pathways and genes previously associated with CAC [134].

Two particularly well-studied examples of disease-related variants are described in the following. First, the secretion of IL-6 by the presence of pro-inflammatory M1-type macrophages was found sufficient to induce polarized Caco-2 intestinal cells to increase their expression of RAC1B through alternative splicing [135]. RAC1B results from the inclusion of an additional exon [136] and is translated into a splicing variant of RAC1, a member of the Rho GTPase family that regulates various cellular processes including cell migration, proliferation, and survival. RAC1B overexpression was observed in samples from IBD patients, as well as in acute colitis triggered in DSS-treated mice [137], but it also characterizes a subgroup of colorectal tumours [138]. An increased expression of RAC1B has been implicated in various tumorigenic processes [139], including senescence evasion [140], enhanced colorectal cell survival and resistance to apoptosis [138,141], the promotion of Apc-dependent tumorigenesis [142], and chemotherapy resistance [143]. Depending on the tumour type, a role in either promoting [144,145] or preventing the epithelial–mesenchymal transition (EMT) [146,147] has also been described.

Mechanistically, the RAC1B protein assumes a constitutively active conformation in cells and activates NF-κB signalling [127,143], which is known to be upregulated in the context of persistent inflammation, and drives cells towards transformation, as described above.

In addition, RAC1B with its additional exon 3b can exacerbate WNT signalling by stabilizing β-catenin. Direct WNT pathway mutations are less common in CAC, but the overexpression of RAC1B can promote the nuclear translocation of β-catenin and facilitate the transcription of proliferation and cell survival genes encoding c-Myc and Cyclin D1 [148]. This has also been observed in a mouse model of induced RAC1B overexpression [142]. Furthermore, RAC1B overexpression has been shown to promote complex formation with NADPH oxidase and increase the production of mutagenic reactive oxygen species (ROS) [149]. The combination of increased NF-κB and WNT signalling with higher mutational load can be expected to significantly increase the risk of progression from chronic inflammation to dysplasia and CAC.

Another gene example with an alternative splicing variant upregulated in IBD is *TACR1* encoding the neurokinin receptor 1 (NK-1R), a G-protein-coupled receptor found in the central, but also the peripheral, nervous system, including the human gastrointestinal tract. When NK-1R is stimulated by its ligand, also known as the pro-inflammatory neuropeptide substance P, it can generate various second messengers and activate the ERK pathway [150]. NK-1R plays modulatory roles in immune responses and has been implicated in the etiology of IBD [151,152,153] and various types of cancer [154,155].

A C-terminally truncated splicing variant, tr-NK-1R, was identified that results from retention of the last intron, leading to a pre-mature stop codon, so the 96 C-terminal amino acids are lacking [156,157]. This truncated NK1R splicing variant differs in ligand affinity and signalling properties and lacks the cytoplasmic tail necessary for receptor desensitization [158,159,160]. tr-NK-1R expression is progressively upregulated in colon samples from patients with IBD, high-grade dysplasia, and CAC [130], suggesting an essential role for this truncated splicing variant in the progression to CAC. Also in sporadic CRC, an increase in tr-NK-1R protein was observed during the adenoma–carcinoma progression, underlining the variant’s role in the progression of colorectal tumours [161].

### 7.2. Splicing Factors Associated with CAC

As described above, the regulation of alternative splicing is achieved through a combinatorial mode of splicing factor binding during exon definition. Accordingly, any change in the expression or activation of a given splicing factor will lead to changes in a variety of splicing events. Various examples of altered levels of CAC-related splicing factors have been studied in more detail and corroborate the fact that they cause significant changes in splicing profiles. In addition, several of the signalling pathways that are dysregulated in IBD and CRC are involved in post-translational modifications of splicing factors, which are indispensable for their normal function in splicing (reviewed in [162]), but also involved in the generation of tumour-associated alternative spliced variants when dysregulated, as detailed in the following.

First, an at least 2-fold increase in expression of SRSF1 was detected in several tumour types, including in about 25% of colorectal tumours [129]. In colon cells, SRSF1 plays a key role in promoting the expression of the above-described splicing variants RAC1B [163] and FADD20 [120] (see Table 1). When present in the cell nucleus, SRSF1 directly binds to alternative target exons and promotes their inclusion into the mature mRNA. This requires sustained SRSF1 phosphorylation by the serine/threonine kinase Serine/Arginine-rich Protein-specific Kinase 1 (SRPK1) [164,165], so that SRSF1 can move into or remain in the nucleus. Although incompletely understood, there is evidence that inflammatory signals can trigger this phosphorylation event in colon cells [109]. In other cases, overexpression of the scaffold protein FAM138B in CRC recruits SRSF1 and SRPK1 into a protein complex, promoting SRSF1 phosphorylation [120]. A bioinformatic analysis suggested that SRSF1 regulates the splicing of 2678 genes in the colon, of which 468 genes were identified as colorectal cancer-related genes [166]. These numbers include SRSF1-dependent splicing in immune cells present in the colon [167,168]. Another study identified 1225 SRSF1-regulated alternative splicing events in the CRC cohort of the TCGA database [120]. Notably, upregulated AKT or ERK activities, common IBD- and CAC-associated events downstream of IL-6 signalling [60,61], also contribute to splicing factor regulation. AKT, for example, promotes SRPK autophosphorylation, resulting in a switch of association from Hsp70- to Hsp90-containing protein complexes, leading to the nuclear translocation of SRPK and enhanced SR protein phosphorylation [169]. Also, ERK-dependent phosphorylation of the splicing factor SRC-associated in mitosis of 68 kDa (SAM68) interferes with its ability to interact with splicing factor U2AF65, thus enhancing recognition and inclusion of the CD44 v5 exon [170,171,172]. The modulation of SAM68 further promotes the pyruvate kinase splicing variant PKM2 involved in the metabolic switch from oxidative phosphorylation to glycolysis in colon tumour cells [126].

Second, the epithelial splicing regulatory protein 1 (ESRP1) expression was found reduced in samples from IBD patients, and lower ESRP1 levels were associated with impaired intestinal barrier integrity and an increased susceptibility to colitis and CAC [122]. In this context, ESRP1 deficiency leads to abnormal splicing in a variety of genes, including CD44 and the G-protein-coupled receptor 137 (GPR137). In the latter, exon 4 inclusion led to the expression of the long GPR137 isoform 2, which stabilizes β-catenin, enabling its nuclear translocation and activation of genes involved in cell proliferation and survival.

Curiously, the stabilization of β-catenin can also be mediated by splicing variant RAC1B, which has been described to be co-regulated by ESRP1 in colorectal cells; however, the upregulation of ESRP1 has been reported in the case of sporadic CRC [173]. Thus, two different ESRP-1-mediated alternative splicing events can lead to WNT pathway activation, and these seemingly contradictory data on the required ESRP1 expression levels may indicate the existence of distinct subtypes of CAC.

Third, the RNA recognition motif-containing protein, RNA-binding protein 47 (RBM47), was found to modify intestinal homeostasis, as mice with the intestinal-specific deletion of *rbm47* showed WNT-pathway activation and the upregulation of stem cell markers with increased cell proliferation and spontaneous tumorigenesis [174]. Consistently, RBM47 mRNA expression was decreased in human colorectal cancer versus paired normal tissue. One directly related alternative splicing event regulated by RBM47 is the inclusion of exon 20 into the tight junction protein 1 (TJP-1, ZO1) transcript. Thus, alternative splicing due to decreased RBM47 expression reduces cell adhesion, favours EMT, and presumably also stimulates proliferation via the ZO-1-regulated nuclear translocation of transcription factor ZONAB [131,174]. Consistently, primary CRC samples showed a decrease in RBM47 gene transcription, which was identified to result from inflammation-driven IL-6/STAT3 signalling [175].

Similarly, a recent publication identified the U2-snRNP component SF3B3 in colon cancer samples and found its overexpression to promote changes in the alternative splicing of specific genes, including the mTOR protein kinase. The respective inclusion of exon 8 into the MTOR transcript increased its signalling to stimulate the proliferation and invasive properties of colorectal cells [125]. The involvement of SF3B3 overexpression in CAC remains to be confirmed.

Fourth, the role of alternative splicing in increasing the risk of malignancy from colitis to CAC extends beyond the context of epithelial cells. One example is the splicing factor elongation factor Tu GTP binding domain-containing 2 (Eftud2), a component of the U5 snRNP that was found to be overexpressed in both colonic epithelial tissue and infiltrating macrophages in a CAC mouse model [176]. Remarkably, knocking out Eftud2 specifically in myeloid cells was linked to a reduction in the levels of inflammatory cytokines, including IL-6. The observed Eftud2 overexpression mediates the alternative splicing of components of the Toll-like receptor-4 (TLR4)-MyoD88 pathway required for the activation of NF-κB signalling, which promotes both chronic intestinal inflammation and tumour progression. A recent review described further IBD- and CAC-related alternative splicing events in immune cells and fibroblasts [177].

Recent studies have further highlighted that complex changes in alternative splicing can depend on epigenetic regulators. Heterochromatin Protein 1γ (HP1γ), for example, recognizes gene-silencing histone modifications, and its reduced expression in UC patients correlated with a higher usage of cryptic splice sites in numerous genes relevant in gut biology. Upon experimental inactivation of the HP1y gene in the mouse gut epithelium, IBD-like traits and extensive changes in alternative splicing [178] were observed. In addition, a reduction in intragenic DNA methylation impaired binding of the chromatin factors MBD1/2/3 and HP1γ, allowing for higher RNA polymerase elongation rates and widespread changes in alternative splicing, including increased CD44 variant exon skipping and enhanced EMT in CRC cells [179].

Future transcriptomic studies will have to take into account the cell types present in collected tissue samples, because epithelial, stromal, and immune cells can contribute to the observed differences in splicing variant profiles. More detailed knowledge of the alternative splicing changes in each cell type, as evidenced recently in samples from young patients with ulcerative colitis and Crohn’s disease [110], may provide new clinically relevant tools.

## 8. Conclusions 

The reviewed data indicate that alternative splicing contributes to the increased colon cancer risk observed in patients with long-term UC, leading to the activation of key signalling pathways involved in tumorigenic transformation in the colon. Key splicing events described so far include RAC1B and NK-1R, but also changes in ESRP1 splicing factor expression that result in a more complex pattern of multiple alternative splicing events. Exploring these events in more detail may thus yield novel biomarkers for disease progression [180]. The data also highlight the pressing need for further research to elucidate the molecular mechanisms through which inflammation-induced alternative splicing drives CAC pathogenesis. For example, recent studies have concluded that transcriptome changes observed in intestinal cells when exposed to inflammatory stimulation can compromise epithelial integrity and cell differentiation and change metabolic pathways and immune responses [181,182]. In one particular study, the transcriptome of 11 patients with long-duration (≥20 years) and 21 patients with short-duration (≤5 years) UC were investigated, in order to understand the transcriptomic differences associated with the duration of UC disease and their association with CAC. Both patient groups differed in 640 genes that were differentially expressed and in 3560 genes exhibiting changes in their alterative splicing pattern [134]. Altogether, the recent transcriptomic studies strongly suggest that these examples are only the very tip of a larger iceberg of alternative splicing changes that are involved in the progression from chronic inflammation to cancer.

A better understanding is required to unravel the potential of targeting alternative splicing as a therapeutic strategy, either by inhibiting specific oncogenic splicing variants (for example, through antisense oligonucleotides) or by attenuating changes in a network of splicing variants dependent on a given splicing factor (for example, through small-molecule compounds binding the splicing factor or inhibiting upstream regulatory signalling pathways). The local delivery of such therapeutics directly to the intestinal tract should be feasible. Such efforts should also help identify downstream effectors of IL-6 signalling, so the pathologic effects of IL-6 can be blocked without compromising intestinal integrity. An interesting example of a specific splicing event that has been explored for therapeutic options is the one generating a soluble GP130 co-receptor (sgp130), which quenches circulating IL-6 and thus downregulates the IL-6 trans-signalling in inflammatory conditions [183]. Distinct alternative sgp130 splicing events were found to be stimulated by the natural compound epigallocatechin gallate from green tea leaves [184] and have led to the development of a therapeutic soluble sgp130 protein fused to the Fc portion of a human IgG1 antibody, which has entered clinical trials [57,185].

## Figures and Tables

**Figure 1 cancers-17-00219-f001:**
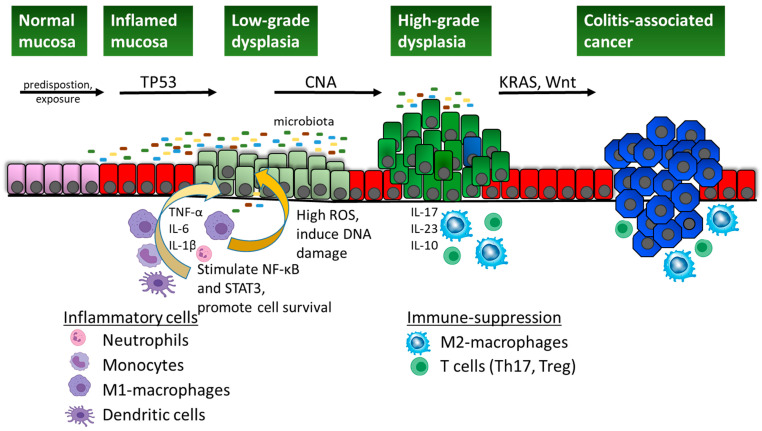
Schematic representation of the dysplasia-to-carcinoma sequence leading to colitis-associated cancer. Following initial colon inflammation (red cell colour), either as a result of environmental exposure and/or genetic predisposition of the individual, immune cells are recruited to the intestinal mucosa. Dendritic cells are antigen-presenting cells that recognize bacteria or cell debris and secrete interferons to activate tissue-resident macrophages and recruit circulating monocytes. These differentiate into M1 macrophages with pathogen-killing abilities that secrete pro-inflammatory cytokines, particularly tumour necrosis factor alpha (TNF-α), interleukin (IL)-6, and IL-1β. These cytokines attract neutrophils for tissue remodelling and pathogen clearance, involving the generation of high amounts of reactive oxygen species. This results in an inflamed mucosa that can become chronic and lead to the development of inflammatory bowel disease (IBD). The secreted cytokines also promote cell survival in the epithelial cells. In addition, the infiltrated immune cells produce reactive oxygen species, which can damage the DNA in epithelial cells and cause genetic mutations. The presence of pathogenic microbes in the gut lumen can further perpetuate the inflamed state and provide carcinogenesis-promoting toxins. Gain-of-function missense mutations in the *TP53* tumour suppressor gene are frequently acquired when low-grade dysplastic mucosa develops (light green cell colour). Subsequently, copy number alterations (CNAs), notably chromosomal abnormalities, develop in the transition from low- to high-grade dysplasia (dark green cell colour). Chronic inflammation also selects for an immunosuppressive microenvironment that is characterized by high levels of IL-17, IL-23, and IL-10 and favours the presence of pro-tumorigenic M2 macrophages and T regulatory (Tregs) or T helper (Th17) cells. Cancer-initiating cells can develop through additional mutations in the KRAS and/or WNT-pathway genes (APC or RNF43), and then progress to CAC (blue cell colour) as they are protected in the immunosuppressive microenvironment.

**Figure 2 cancers-17-00219-f002:**
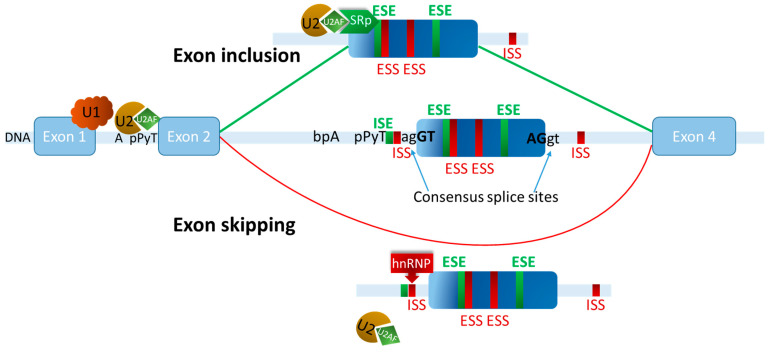
Exon definition mediates the regulation of alternative splicing by sequence elements and their binding factors. In the rare case of a short intron, shown here between exons 1 and 2, direct interactions between the spliceosome components bound to the 5′ and 3′ splice sites can mediate correct splicing. In most human genes, however, short exons are flanked by much longer introns so that efficient splicing, as well as the regulation of alternative splicing (shown here for exon 3), require preceding exon definition. For this, short splice-enhancing (green colour) or -silencing (red colour) sequence elements are present in the exon or the adjacent intronic regions. These are recognized by splicing factors such as members of the SR or hnRNP protein families, which can then either promote (top) or prevent (bottom) spliceosome assembly on the alternative exon. In addition, the consensus splice site sequences can have a stronger or weaker binding affinity for the core-spliceosome components. In consequence, the alternative exon 3 is either included (Exon inclusion, green lines, top) or excluded (Exon skipping, red line, bottom) from the mature mRNA. Alternative splicing can be modulated in a combinatorial manner by synergistic or antagonistic splicing factors, the outcome being determined either by their relative (often tissue-specific) differences in expression levels, or their activation or subcellular location due to post-translational modifications in response to cellular signals. U1 = U1 small nuclear ribonucleoprotein particle (snRNP), U2 = U2 snRNP, U2AF = U2 snRNP auxiliary factors, pPyT = polypyrimidine tract, bpA = branch point adenine, ESS/ISS = exonic/intronic splice silencer, ESE/IES = exonic/intronic splice enhancer, SRp = SR-family protein, hnRNP = heterogeneous nuclear ribonucleoprotein family member.

**Table 1 cancers-17-00219-t001:** Examples of well-characterized alternative splicing-derived proteins expressed in sporadic or colitis-associated colorectal cancer.

Gene Name	Colon Cancer-related Variant	Splicing Event	Functional Properties	Ref.
*BCL2L1*	BCL-xL	Alternative 5′ splice site usage in exon 2	Anti-apoptotic effect at mitochondria	[113,114]
*CASP 9*	Caspase 9b	Skipping of exons 3–6	Anti-apoptotic effect in caspase cascade	[115]
*CCND1*	CCND1b	Intron 4 retention creates alternative C-terminus	Cyclin D variant for cell cycle progression	[116]
*CD44*	CD44 v4–10	Inclusion of variable exons 4–10	Increased interaction with HGF stimulates proliferation	[117]
	CD44 v6	Inclusion of exon v6	Enhances EMT, cell motility, and invasion	[118]
*CEACAM1*	CEACAM1-S	Skipping of exon 7	Promotes migration and invasion	[119]
*FAAD20*	FAAD20-t	Exon 4 inclusion	Enhanced binding to Fanconi anemia group A protein improves DNA repair	[120]
*FAS (CD95)*	sFAS	Skipping of exon 6	Anti-apoptotic soluble Fas isoform	[121]
*GPR137*	GPR137-L	Exon 4 inclusion	WNT signalling activation	[122]
*IL6R*	sIL-6R	Skipping of exon 10	Lacks transmembrane domain and is secreted as soluble protein	[55]
*ITGA6*	ITGA6A	Skipping of exon 25	Integrin α6 variant that activates the WNT/β-catenin pathway	[123]
*MKNK2*	MNK2b	Skipping of exon 14a	Pro-oncogenic variant lacking a domain capable of activating p38α–MAPK-mediated cell death	[124]
*MTOR*	fl-mTOR	Exon 8 inclusion	Increased pro-proliferative signalling	[125]
*PKM*	PKM2	Inclusion of exon 10 instead of exon 9	Increased aerobic glycolysis and lactate generation for metabolic adaptation	[126]
*RAC1*	RAC1B	Inclusion of exon 3b	Promotes NF-κB signalling and cell survival	[127]
*RON (MST1R)*	deltaRON	Skipping of exon 11	Enhances EMT, cell motility, and invasion	[128]
*RPS6KB1*	S6K-p31 (isoform 2)	Alternative C-terminal cassette exons (6a, 6b, and 7a)	S6-kinase variant activates mTORC1 activity and cell growth	[129]
*TACE1*	tr-NK-1R	Intron 4 retention	Different ligand affinity and signalling outcomes	[130]
*TJP1 (ZO1)*	TJP1-E20	Skipping of exon 20	Promotes EMT and cell proliferation	[131]
*VEGF*	VEGF165a	Proximal splice site usage in exon 8	Promotes angiogenesis	[132]
*UPF3A*	UPF3A-S	Skipping of exon 4	Promotes proliferation	[133]

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
