# Peer review of "Alternative Splicing at the Crossroad of Inflammatory Bowel Diseases and Colitis-Associated Colon Cancer"

_cancers, 2025, doi:10.3390/cancers17020219_

Round 1

Reviewer 1 Report

Comments and Suggestions for Authors

This review focuses on inflammatory bowel diseases (IBDs) and their associated risk of cancer development. The authors provide a concise overview of key pathways implicated in inflammation-driven tumorigenesis, including the NF-kB, interleukin, and Wnt signaling pathways. They emphasise the critical role of alternative splicing regulation in disrupting gene expression, which may contribute significantly to the development of colon-associated cancers (CAC). The review concludes by underlining the need for further transcriptomic studies to elucidate alternative splicing programs that potentially drive CAC pathogenesis

The review is well-written and informative, reflecting a good standard of quality.

However, the discussion could benefit from greater clarity regarding how alternative splicing regulation mechanisms, well-established in tissues other than intestinal cells, may also be relevant to CAC development if the requisite factors are expressed. For instance, signaling pathways that regulate the phosphorylation of various splicing factors, as illustrated in the review with SRSF1, are likely to significantly influence the alternative splicing of their target genes. Expanding on this point could broaden the reader's interest beyond SRSF1 to other splicing factors and provide additional depth and context to this important aspect of the review. For further insights, see the recent comprehensive review by Kretova et al., 2023 (eLife).

Another important mechanism of alternative splicing regulation that the authors have overlooked is the role of co-transcriptionality and the influence of epigenetic and chromatin modifications (see Zhang et al., 2020, *Frontiers in Genetics*). While this is briefly mentioned in lines 325–329 (ref. 102), the topic warrants more detailed exploration due to its relevance to the review's focus.

The interplay between alternative splicing regulation and chromatin factors is increasingly recognized as significant, particularly in the context of inflammatory bowel diseases (IBD) and associated cancer risks. For example, Mata-Garrido et al. (2022) demonstrated the role of HP1γ/CBX3 in modulating RNA splicing through the control of splicing noise and the suppression of cryptic splice sites. This regulation has been observed in genes with functions critical to gut biology, including PKM, CDH1, and CD44, which are cited in this review as being differentially spliced during CAC development. Notably, similar to CD44 alternative splicing (Saint-André et al., 2011), the alternative splicing of VEGF and other genes is dependent on H3K9me2/3 and HP1γ (Salton et al., 2014), suggesting a potential link with TGF-β signaling via co-transcriptional mechanisms.

Beyond HP1γ/CBX3, other chromatin-associated proteins also influence alternative splicing and may impact IBD and colon cancer progression. DNA methylation, for instance, directly and indirectly influences alternative splicing by modulating RNA polymerase II (RNAPII) elongation speed and chromatin structure within the gene body, as well as by altering the expression of splicing factors during the epithelial-to-mesenchymal transition (EMT) in tumoral colon cells (Batsché et al., 2021).

Additionally, genes like FGFR2, known for their roles in EMT, have been shown to be influenced by changes in the epigenetic landscape (Luco et al., 2010).

For instance in colon cancer, mutations in SETD2 disrupt splicing of tumor suppressor genes (see the study titled "Histone methyltransferase SETD2 modulates alternative splicing to inhibit intestinal tumorigenesis," published in the Journal of Clinical Investigation in 2017, by Huairui et al.). This research demonstrates that SETD2, which regulates H3K36me3, plays a crucial role in alternative splicing. Its inactivation leads to aberrant splicing events that contribute to colorectal cancer progression. Specifically, the loss of SETD2 function results in the accumulation of dishevelled segment polarity protein 2 (DVL2) mRNA through nonsense-mediated decay–coupled alternative splicing regulation, thereby augmenting Wnt signaling (PMID: 28825595).

Highlighting these mechanisms would add valuable context to the review and underscore the importance of chromatin dynamics in splicing regulation and disease progression.

Kretova, M.; Selicky, T.; Cipakova, I.; Cipak, L. Regulation of Pre-mRNA Splicing: Indispensable Role of Post-Translational Modifications of Splicing Factors. Life 2023, 13, 604. https://doi.org/10.3390/life13030604

Zhang et al., “The Crosstalk Between Epigenetic Mechanisms and Alternative RNA Processing Regulation,” Frontiers in Genetics 11 (2020): 998, https://doi.org/10.3389/fgene.2020.00998.

Mata-Garrido J, et al. 2022. The Heterochromatin protein 1 is a regulator in RNA splicing precision deficient in ulcerative colitis. Nat Commun. 2022 Nov 18;13(1):6834. doi: 10.1038/s41467-022-34556-3.

Saint-Andre et al., “Histone H3 Lysine 9 Trimethylation and HP1γ Favor Inclusion of Alternative Exons,” Nature Structural & Molecular Biology 18, no. 3: 337–44, https://doi.org/10.1038/nsmb.1995;

Batsche et al., “CD44 Alternative Splicing Senses Intragenic DNA Methylation in Tumors via Direct and Indirect Mechanisms,” Nucleic Acids Research 49, no. 11: 6213–37, https://doi.org/10.1093/nar/gkab437.

Reini F. Luco et al.Regulation of Alternative Splicing by Histone Modifications.Science327,996-1000(2010).DOI:10.1126/science.1184208

M. Salton, T. C. Voss, and T. Misteli, “Identification by High-Throughput Imaging of the Histone Methyltransferase EHMT2 as an Epigenetic Regulator of VEGFA Alternative Splicing,” Nucleic Acids Research, November 20, 2014, https://doi.org/10.1093/nar/gku1226.

Reviewer 2 Report

Comments and Suggestions for Authors

 The review of Matos et al. is devoted to data on alternative splicing in inflammatory bowel disease and colitis-associated colorectal cancer.

This is a very interesting review covering parallels and differences in signaling, mutations in both sporadic and CAC CRC, as well as well-characterized alternative splicing derived proteins and splicing factors. The review is well written and full of important scientific information. It is well organized and has a certain logical structure; it contains very clear, relevant and comprehensive figures.

Minor terms are:

1) 147...149. “This explains why clinical trials with anti-IL-6 antibody therapies revealed limited benefits and several patients exhibited adverse gastrointestinal effects”

How does it explain? The meaning is missing.

2) 194. Methylation of what?

3) 182. What is inactive UC?

4) Figure 1. M1 macrophages are widely recognized as antitumor. However, they actually confer proinflammatory properties in colitis that many readers know much less about. Could the authors please briefly explain the inflammatory properties in colitis of all the cell types shown, including M1 macrophages?

Reviewer 3 Report

Comments and Suggestions for Authors

In this review titled "Alternative Splicing at the Crossroads of Inflammatory Bowel Disease and Colitis-Associated Colon Cancer," the authors describe several transcriptomic studies to elucidate the molecular mechanisms by which inflammation-driven alternative splicing drives the pathogenesis of colitis-associated colon cancer.

The topic is obviously interesting and several research articles are elucidating the molecular mechanisms, however very recently some reviews very similar to this one proposed by the two authors have been published.

References:

- Zhou J, Zhang Q, Zhao Y, Song Y, Leng Y, Chen M, Zhou S, Wang Z. The regulatory role of alternative splicing in inflammatory bowel disease. Front Immunol. 2023 Apr 21;14:1095267. doi: 10.3389/fimmu.2023.1095267.

- Zou C, Zan X, Jia Z, Zheng L, Gu Y, Liu F, Han Y, Xu C, Wu A, Zhi Q. Crosstalk between alternative splicing and inflammatory bowel disease: Basic mechanisms, biotechnological progresses and future perspectives. Clin Transl Med. 2023 Nov;13(11):e1479. doi: 10.1002/ctm2.1479.

-  Li, D., Tan, Y. Dysregulation of alternative splicing is associated with the pathogenesis of ulcerative colitis. BioMed Eng OnLine 20, 121 (2021). https://doi.org/10.1186/s12938-021-00959-4

The research papers published in the last year are not many and not precisely focused on this topic, so the publication of the present reviewt shows an overall low priority of acceptance.

In particular, the description of multi-step tumorigenesis of colorectal cancer development and malignant progression summarized in Figure 1 is approximate and incorrect.

The acquisition of the combination of mutant p53 with other driver mutations during multi-step tumorigenesis (APC, KRAS, PIK3CA, SMAD4, and TP53) is inaccurate. The latest transcriptomic studies on transgenic mice and patient-derived organoids have elucidated the different mechanisms of acquisition of GOF mutant p53 and subsequent LOH of the wild type p53 allele in CRC and in the inflammatory phase of colitis (doi: 10.1093/jmcb/mjy075; https://doi.org/10.1038/onc.2017.194).

Overall, the manuscript does not discuss innovative aspects and is confounding in some mechanisms.

Reviewer 4 Report

Comments and Suggestions for Authors

Review on the manuscript titled :” Alternative splicing at the crossroad of inflammatory bowel  diseases and colitis-associated colon cancer” by Matos, Jordan

                The authors reviewed the inflammatory bowel diseases (IBD) which majorly manifested by ulcerative colitis (UC) or Crohn’s disease (CD). Notably, the authors underline the observation that UC may lead to colitis-associated colon cancer (CAC), in contrary to a standard colorectal cancer (CRC). In their review, they essentially stress on Alternative splicing (AS) impact details and features.

                The manuscript contains 8 chapters:

1.       Inflammatory Bowel Disease

2.       Cancer risk in patients with Inflammatory Bowel Disease

3.       Specificities of colitis-associated cancer

4.       Molecular pathways in colitis-associated cancer, including:

4.1.  NF-κB activation

4.2.  IL-6/STAT3 signalling

4.3.  Wnt pathway

4.4.  Other pathways

5.       Transcriptomic reprogramming through alternative splicing

6.       Exon definition governs alternative splicing decisions

7.       The role of alternative splicing in CAC development, including:

7.1.  Splicing variants associated with CAC

7.2.  Splicing factors associated with CAC

8.       Conclusion and Outlook

As it looks, the key idea/mission of the review is expressed in Conclusion: “The reviewed data indicate that alternative splicing can significantly contribute to the increased colon cancer risk observed in patients with long-term UC, leading to the activation of key signaling pathways involved in tumorigenic transformation in the colon.”

In the course of the review, the authors commit essential efforts in compiling the studies in the area, and present notable facts and info on drug targets. Still, their work is quite dangerously close to (Zhang et al., 2023) they reference. While they compile Table 1, as well as typical cancer mediated immune response pathways in chapter 4, it is not quite clear what is the specific aim of the manuscript. If they are interested on UC impact on CAC, they should closely pursue the specific effects UC elicited on cancer state. While they correspond that UC just invokes non-specific inflammatory response, there is small room for any contemplation, or specifics. So, I’d be grateful to see ANY direct splicing-related specifics of UC impact(s) on CAC to justify the paper’s title. Otherwise, it is just slightly modified Zhang et al., 2023 paper.

Some notes are listed below:

1)      Table 1 should be referenced accordingly (same as in Zhang et al., 2023).

2)      It’s advised some concise declarations in conclusion section on specific impact UC exemplifies in cancer etiology in terms of splicing events. There are some relevant splicing observations in chapter 7, but I believe they aren’t detailed explicitly enough.

3)      The presentation of basics of alternative splicing is a little bit out of frame. I consider chapter 6 a bit redundant, since no further real explicit splicing events/details are outlined.

4)      It’s probably worthwhile to mention (in introduction) that splicing –oriented drug targets (see Zhang et al., 2023) are quite suitable for intestinal tract since they would be delivered in the right place, though I believe they may also access to the blood as well.

Reviewer 5 Report

Comments and Suggestions for Authors

Matos and Jordan have reviewed inflammatory bowel diseases and colitis-associated colon cancer, focusing on the association of splicing variants of genes with these diseases. The review is thorough, and key players in cancer are effectively highlighted. While the text is well-written, it may benefit from some minor adjustments. Below are some suggestions that I hope the authors find useful.

Major:

Suggestion: missense mutation and their occurrences reviewed in 97-107, may be presented in a barchart which shows frequency of missense mutations in TP53 in CRC and CAC; a side barchart showing the frequencies (CRC ---|-- CAC).

4.3 section, information in this section does not present very strong evidence compared to other highlighted pathways. I suggest either authors provide more detailed information on the mechanism of action or exclude this part.     

375- 384: Although the alternative splice (AS) event of RAC1B in table1 has been highlighted, in the text it is not very clear whether the referred overexpression of gene is with inclusion exon 3b?

Conclusion and outlook section: reviewed examples in the manuscript although provide evidence on contribution of AS but “significant contribution of alternative splicing”, I think is an overstatement without providing a measurable metric in relation to other mutations identified to be associated with CAC and CRC. For example, if you look at available data in TCGA, in relation to point mutations, CNAs, other SVs, and AS what is the fraction of identified AS in the cases?

Minor:

36: “The risk of developing CRC is about 2 fold-increased compared to the general population in ???, especially”  ?

37-38: Please rephrase, it is a bit difficult to grasp the main message

51: crypt basis base

76-80: improve the text, rather long sentence with a lot of information. 

Round 2

Reviewer 1 Report

Comments and Suggestions for Authors

The authors have satisfactorily incorporated all the suggestions I provided, and the manuscript now meets the required standards without the need for further modifications.
I confirm that the revised version of the manuscript is suitable for publication.

Reviewer 3 Report

Comments and Suggestions for Authors

The manuscript is improved both in terms of the discussion of molecular mechanisms and the type of organization of the contents.

It is not far from what has been recently published in the literature, but what is discussed by the authors is correct and clearer.

Overall the present form of the manuscript is acceptable.